# Stylometry can reveal artificial intelligence authorship, but humans struggle: A comparison of human and seven large language models in Japanese

Wataru Zaitsu[1¤*], Mingzhe Jin[2], Shunichi Ishihara[3], Satoru Tsuge[4], Mitsuyuki Inaba[5]

**1** Faculty of Psychology, Mejiro University, Tokyo, Japan, **2** Institute of Interdisciplinary Research, Kyoto University of Advanced Science, Kyoto, Japan, **3** Speech and Language Laboratory, Australian National University, Canberra, Australia, **4** School of Informatics, Daido University, Aichi, Japan, **5** College of Policy Science, Ritsumeikan University, Kyoto, Japan

¤ Current Address: Department of Psychological Counselling, Faculty of Psychology, Mejiro University, 4-31-1 Nakaochiai, Shinjuku-ku, Tokyo, Japan
* w.zaitsu@mejiro.ac.jp

## Abstract

The purpose of this study was to estimate the artificial intelligence (AI) detection potential using stylometric analysis in Study 1 and examine the AI detection abilities of humans in Study 2. In Study 1, we compared 100 human-written public comments with 350 texts generated by seven large language models (LLMs) (ChatGPT [GPT-4o and o1], Claude3.5, Gemini, Microsoft Copilot, Llama3.1, and Perplexity) using multidimensional scaling (MDS) to visualize differences by focusing on three stylometric features (phrase patterns, part-of-speech bigrams, and unigrams of function words). In general, each stylometric feature can distinguish between LLM-generated and human-written texts. In particular, three integrated stylometric features achieved perfect discrimination on MDS dimensions. Interestingly, only Llama3.1 exhibited distinct characteristics compared with the other six LLMs. The random forest classifier also achieved 99.8% accuracy. In Study 2, we performed an online survey to assess the Japanese participants' AI detection abilities by presenting LLM-generated and human-written texts, as used in Study 1. 403 participants tackled "AI or Human" judgment task and estimated their own confidence, revealing that overall human AI-detection ability was limited. Moreover, in our materials, more advanced ChatGPT(o1), plausibly reflecting relatively greater fluency and polish, tends to mislead the participants to believe "human-written" texts compared with ChatGPT(GPT-4o) and improves their confidence for their own judgments. Furthermore, an additional comment from the survey suggested that participants primarily relied on superficial impressions based on phraseology, expression, the ends of words, conjunctions, and punctuation marks in judgments. These findings have important implications for various scenarios, including public policy, education,

**Data availability statement:** All relevant data are included within the Supporting Information files.

**Funding:** This work was partially supported by JSPS KAKENHI (grant number: JP23K11107). The funders had no role in the study design, data collection, analysis, and decision to publish, except for the Publication Fee.

**Competing interests:** We have declared that no competing interests exist.

and marketing, where the rapid and reliable detection of AI-generated content is increasing.

## Introduction

With the release of ChatGPT [1] by OpenAI on November 30, 2022, generative artificial intelligence (AI), including large language models (LLMs), has rapidly spread worldwide and the development of generative AI has intensified globally. These movements have bestowed benefits to humans but occasionally pose hazards. It has been highlighted that disinformation generated by AI such as fake news causes various risk such as bank runs [2], election influence [3], and legal system challenges [4]. Unfortunately, these AI-generated contents, including texts, are already human-like, and several previous findings [5–8] have suggested limitations of human abilities to distinguish between LLM-generated and human-written English texts. Incidentally, no studies have addressed this issue in Japan. In contrast to the detection disabilities of humans, various researchers have attempted to distinguish between LLM-generated and human-written texts using classical machine learning (ML) [9–13] or deep learning (DL) [14–15]. In a previous study, Desaire et al. [9] successfully classified ChatGPT-3.5 and human-written academic papers using XGBoost (over 99% accuracy). Zaitsu & Jin [10] targeted the Japanese language and reported approximately 100% accuracy in distinguishing academic papers generated by AI (ChatGPT 3.5 and 4.0) from those written by humans using random forest (RF). In addition, Zaitsu et al. [11] compared the detection accuracies of public comments generated by zero- and one-shot prompting using ChatGPT 3.5 and 4.0. They concluded that texts generated by one-shot prompting posed challenges for analysts, dropping to approximately 90%. Alamleh et al. [12] attempted to use various ML algorithms (logistic regression, decision trees, support vector machines, neural networks, and RF), but they did not demonstrate high accuracy, ranging from 63% to 77%.

At present, it is likely to be not easily interpretable for humans because it is difficult to determine why several MLs including DL classify a text as AI-generated instead of human-written owing to the opaque, "black-box" nature of the internal processes. Moreover, when using ML, it is difficult to visually grasp the similarities between LLM-generated and human-written texts. Among these previous studies, to the best of our knowledge, none of the previous studies has visually displayed similarities between LLM-generated and human-written texts, except for Zaitsu & Jin [10] and Zaitsu et al. [11] targeting only ChatGPT. The display of similarities gives us the following benefits: (1) Authorship consistency within each LLM or human: Analyzing similarities leads us to grasp a range of intra-individual variations in texts authored by the same LLM or human. (2) Author differences between each LLM and human: We can understand the extent to which the range of texts generated by LLMs differs from those authored by other LLMs or humans in terms of distance. If the range between an LLM and a human does not overlap, this result indicates that complete

discrimination is possible. This study used not only ChatGPT but also multiple LLMs and performed a multivariable analysis to visually display the similarities between LLM-generated and human-written texts.

Thus far, previous studies [9–12] mainly targeted and handled ChatGPT. In recent years, ChatGPT has been rapidly developed and refined, leading to the continuous release of updated versions (e.g., GPT-4o, o1, and o3-mini). This raises several questions: Are these advanced models capable of generating human-like texts? Are more advanced models capable of mimicking the different linguistic levels of human writing styles? Specifically, can they learn not only surface-level characteristics (e.g., the use of characters, words, and phrases) but also deeper, more profound stylistic elements related to higher-level linguistic information (e.g., syntax and pragmatics)? In other words, are texts generated by more advanced AIs more similar to those written by humans in terms of the various stylometric features extracted from the texts? Moreover, is stylometric analysis more challenging for more evolved models than older ones? Although limited to older models (ChatGPT 3.5 and 4.0), a previous study [9] analyzed academic papers using classical multidimensional scaling (MDS) to visually demonstrate the distribution of stylometric features, which exhibited overlap and no differences in the distribution of stylometric features (bigrams of part-of-speech (POS) and postpositional particle words). In contrast to these academic papers, when analyzing public comments with more degrees of freedom than academic papers, different version models (i.e., ChatGPT 3.5 and 4.0) visually exhibited no overlaps and different characteristics for several stylometric features (e.g., phrase patterns and POS bigrams) on the two dimensions of MDS [11]. However, it remains unclear which model closely resembles humans. In addition to ChatGPT, various LLMs have recently emerged worldwide, including Gemini (Google DeepMind), Claude (Anthropic), Perplexity (Perplexity AI), Microsoft Copilot (Microsoft), and Llama (Meta AI). These LLMs have common architectures known as transformers [16], and are pretrained on web data. Consequently, it is expected that the stylistic properties of texts generated by various LLMs, which can be represented via the extracted stylometric features, will resemble one another. However, previous studies have mainly ChatGPT [9–12] as stated before. Therefore, the purpose of the current study (Study 1) is to examine the similarities among various LLM-generated and human-written texts using the following hypotheses: (1) Hypothesis 1: Texts generated by seven recently developed LLMs (GPT-4o, GPT-o1, Claude3.5, Gemini, Microsoft Copilot, Llama3.1, and Perplexity) are different from human-written texts in terms of stylometric features. Therefore, stylometric analysis can distinguish between the two types of texts. (2) Hypothesis 2: LLMs display these similarities. In other words, the range of intra-LLM variation of texts is not different from those of other LLMs, and they overlap with the dimensions of the MDS. (3) Hypothesis 3: Among the LLMs that overlap, texts generated by the more advanced GPT-o1 are more similar to human texts than those generated by the older model (GPT-4o) from the perspective of each stylometric feature. In other words, the distance between GPT-o1 and humans is shorter than that between GPT-4o and humans on the MDS.

Notably, in this study we mainly used MDS because of offering several advantages. First, the MDS faithfully reflects the pairwise distances of the data as distances (or similarities) on the dimensions. Second, MDS ensures high reproducibility, as the same input data always produce the same output, in contrast to t-SNE. Third, the output of MDS (coordinates in a low-dimensional space) is easy to interpret. However, this study supplementally tried to classify both LLMs and Human using RF classifier to verify Hypothesis 1 as same previous studies [10,11]. In belief, this study extended these previous studies by evaluating new text (e.g., public comments) using multiple LLMs and by examining a human-judgment as stated next.

In Study 2, we conducted an online psychological questionnaire survey targeting the non-expert general public to examine individuals' AI detection abilities by presenting LLM-generated and human-written texts used in Study 1 to answer the following two questions: How accurately can Japanese people distinguish whether the Japanese texts were generated by LLMs or written by humans? How confident are they about their assessments? In Study 2, we investigated the relationship between the similarity of stylometric features among the seven AI and humans shown in Study 1 and the individuals' AI detection abilities, including judgment accuracy and confidence level. Several previous studies [5–8] have suggested the difficulty of AI detection by humans. Köbis et al. [5] reported individuals' difficulty in differentiating

poems generated by the older model GPT-2 and positive correlations between detection accuracy and confidence level. The study [6] compared humans' judgments in three different domains (stories, news articles, and recipes) generated by GPT-2 and GPT-3. This result indicated that the more evolved the model, the more humans misjudged the task "AI or human" (Overall accuracy: 0.58 on GPT-2 versus 0.50 on GPT-3). Silva et al. [7] also reported that even experienced editorial board members struggled to distinguish between academic papers on AI and humans, but members with prior LLM experience improved accuracy. In contrast, only Chein et al. [8] reported over chance level accuracies (57% of texts such as news and 78% of comments on social media), and participants with higher fluid intelligence (e.g., vocabulary skills, comprehension ability, and insight) tended to have the ability to correctly distinguish humans from AI. Therefore, humans are expected to have difficulty detecting LLM-generated texts, even in Japanese. Moreover, previous studies did not consider various LLMs, including the more refined and recent ChatGPT, instead of GPT3.5 and GPT4. We can imagine that a more evolved LLM might be more human-like and more easily deceive us, implying that it can make incorrect judgments with higher confidence. Thus, in Study 2, we verified the following hypothesis: Hypothesis 4: The task of detecting AI-generated text in Japanese is challenging. In other words, under the Bayesian framework stated below, it is predicted that the 95% credible interval for the log odds ratio (i.e., the difference between individuals' AI detection accuracy and chance-level performance) will include zero or deviate in the direction of incorrect judgments regardless of the type of LLMs (GPT-4o, GPT-o1, Claude3.5, Gemini, Microsoft Copilot, Llama3.1, and Perplexity). Hypothesis 5: More advanced LLMs (i.e., GPT-o1) tend to increase the difficulty of AI detection compared with older types (i.e.,GPT-4o).

## Study 1

### Sample

In Study 1, we targeted 100 Japanese public comments as human-written texts. These texts were the same as those used by Zaitsu et al. [11], referred to as "HM" texts (mean of 661.3 and *SD* 132.0 of the number of characters). These data were obtained from the e-Gov website (https://www.e-gov.go.jp), published by Japanese national administrative agencies. These public comments included various contents as shown by the following titles; "The Role of Post Offices in the Context of an Aging Society and Population Decline", "Conservation and Management of Living Marine Resources", "Convention on Biological Diversity", and "Market Validation in the Telecommunications Sector".

Apart from the HM texts, 50 texts were generated for each LLM (GPT-4o, GPT-o1, Claude3.5, Gemini, Microsoft Copilot, Llama3.1, and Perplexity) with zero-shot prompts, including the role (e.g., general citizens and business persons) and title of the public comment in December 2024, following the method of Zaitsu et al. [11]. Consequently, 100 human-written texts were compared with 350 LLM-generated texts. The means and standard deviations of the text characters for each LLM were as follows: GPT-4o (922.4 and 100.0), GPT-o1 (888.5 and 114.5), Claude3.5 (710.8 and 120.6), Gemini (912.8 and 140.9), Microsoft Copilot (823.8 and 91.9), Llama3.1 (919.7 and 179.0), and Perplexity (1136.9 and 215.7).

### Japanese stylometric features

Several Japanese stylometric features have demonstrated high accuracy in both authorship analysis [17–19] and AI detection [10,11]. In particular, three stylometric features (function word unigrams, POS bigrams, and phrase patterns) are the most effective for AI detection. Therefore, this study also reused and focused on these stylometric features and counted the frequency of their occurrence in each text. Additionally, these frequency data were divided by the number of all stylometric features in the same text to standardize into relative frequency data and percentages for avoiding dependency against the number of characters in the texts: Concretely, if we confirmed the 37 pieces of post particle words "の" in a text, the value was transformed into 12.5 (i.e., (37/295)*100).

The unigrams (i.e., $N=1$ of $N$-gram) of function words were distinguishable: "の (postpositional particle)", "ない (auxiliary verb)", and "また (conjunction)". These stylometric features have no meaning and tend to be independent of text content. Therefore, the current study also focused on the unigrams of function words, including punctuation marks, such

as Japanese commas and periods. In this study, we used the Japanese POS tagger Mecab [20] to attach each word to the POS tags.

POS bigrams, in the case where $N=2$ of the *N*-gram specialized in POS, are frequently used in authorship analyses [17,18]. In addition, previous studies [10,11] have shown the efficacy of AI detection. The sentence "猫が走る" in Japanese, meaning "cat runs" in English, can be divided into "猫 (noun)", "が (postpositional particle)", and "走る (verb)". Therefore, the POS bigrams correspond to "noun + postpositional particle" and "postpositional particle + verb".

Among the Japanese stylometric features, phrase patterns, whose usefulness in authorship analysis was first demonstrated by Jin [19], also exhibited the highest capability for AI detection [11]. Specifically, focusing only on phrase patterns achieved a recall rate of 95% for texts generated by ChatGPT (zero-shot prompting) and 96.0% for human-written texts. First, we performed a morphological analysis to attach each word to the POS tags. Second, these sentences were divided into phrase units using syntactic analysis with the Japanese parser CaboCha [21]. Finally, we counted the frequency of the combination of function words and POS-masked content words within all phrases: For instance, (1) the Japanese sentence "猫 + が" within a phrase was transformed to "noun + が", the former word means "cat" (i.e., content words) and the latter is a postpositional particle, (2) the sentence "明日 + へ + の" was changed into "noun + へ + の": Only "明日" was masked by POS because of the content word "tomorrow" and both the latter words left alone because of a postpositional particle, and (3) the content word "走る", meaning "run", was masked to symbol "verb".

In addition to the separate analysis of each of the aforementioned stylometric features, this study ensembled all stylometric features to assess their collective discriminating potential between LLM-generated and human-written texts.

## Analysis procedure

To verify Hypothesis 1–3, we performed metric MDS to visualize the stylometric features of each text. In the MDS, this study adopted the symmetric Jensen–Shannon divergence distance ($d_{SJSD}$) [22] as a similarity index among texts on dimensions. Moreover, we arranged all 450 texts and labeled seven LLMs (GPT-4o, GPT-o1, Claude3.5, Gemini, Microsoft Copilot, Llama3.1, and Perplexity) on two MDS dimensions. MDS was performed using the *cmdscale* function of the **stats** package of the R language.

In addition to MDS, to verify Hypothesis 1, we attempted to classify both LLMs and humans by focusing only on integrated stylometric features using an RF classifier and the LOOCV procedure, as in previous studies [10,11]. For RF classification, we used the *randomForest* function of the **randomForest** package of the R language with ntree=1000, keeping all other parameters at their default values.

## Results and discussion in Study 1

According to Fig 1 (unigram of function words), 2 (POS bigrams), 3 (phrase patterns), and 4 (integrated features), the MDS showed that the stylometric features visually differed between LLM-generated and human-written texts. For reference, we supplementarily performed "AI or Human?" task using an RF classifier focusing on integrated stylometric features and obtained the following results: Accuracy (99.8%), recall for LLMs (100.0%), recall for humans (99.0%), precision for LLMs (99.7%), precision for humans (100.0%), F1 for LLMs (99.9%), and F1 for humans (99.5%). In addition, sensitivity analyses, tuned across each hyperparameter (ntree, mtry, maxnodes, and nodesize) together with down-sampling to balance classes from AI=350/Human=100 to AI=100/Human=100, also showed stable performances. These results support Hypothesis 1. Moreover, integrated stylometric features can almost perfectly distinguish AI from humans (Fig 4) compared with single stylometric features (Figs 1–3); therefore, these results indicate that a combination of several features is important for AI detection because of enhanced accuracy (i.e., incremental validity).

Figs 1–4 largely supported Hypothesis 2, but only Llama3.1 among the seven LLMs was placed away from the other six LLMs on the MDS dimensions. This interesting result indicates the difference in Llama3.1 among LLMs in terms of

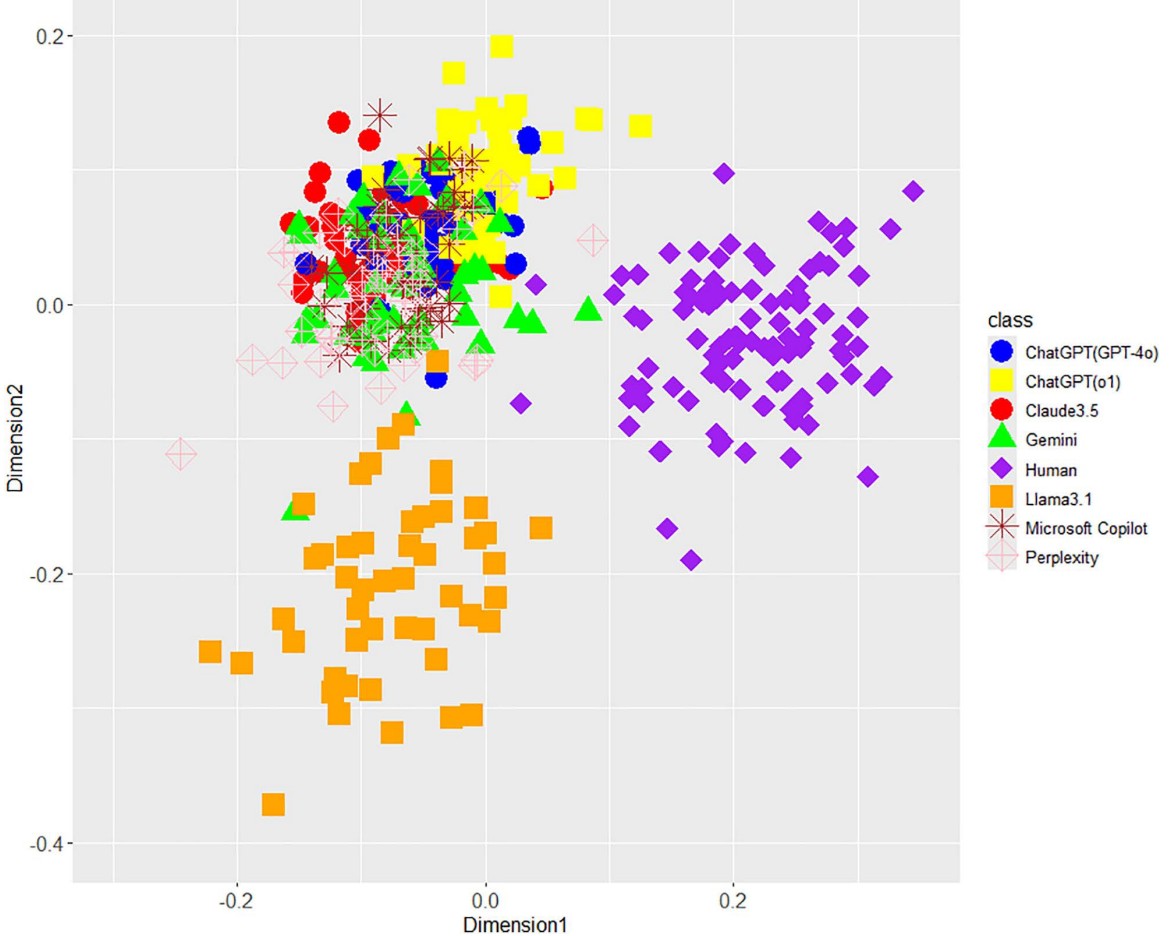

**Fig 1. MDS configuration for function word unigrams in eight classes including each LLM and human.** * The distances (e.g., symmetric Jensen–Shannon distances) between each object on the two dimensions indicate relative similarity of concerned stylometric features.

unigrams of function words, POS bigrams, and phrase patterns. Regarding this point, because the texts generated by Llama3.1 may give us a different impression compared with the other six LLMs, we examined the human judgments and impressions against texts generated by LLMs including Llama3.1 on AI detection tasks and their confidence in Study 2.

Concerning Hypothesis 3, a few of GPT-o1 texts were visually closer to the distribution of human-written texts than those of GPT-4o (Figs 1, 3, and 4), but both GPT-4o and GPT-o1 texts overlapped as a whole. On the contrary, the average distances between GPT-o1 and human were larger instead of the ones of GPT-4o except for phrase patterns, and each effect size wase considerably low: 0.419 (*SD* 0.031) vs 0.425 (*SD* 0.031) of function word unigrams with Cohen's *d* (0.19), 0.458 (*SD* 0.031) vs 0.470 (*SD* 0.032) of POS bigrams with Cohen's *d* (0.04), 0.615 (*SD* 0.027) vs 0.603 (*SD* 0.026) of phrase patterns with Cohen's *d* (0.45), and 0.505 (*SD* 0.024) vs 0.505 (*SD* 0.023) of integrated features with Cohen's *d* (0.00). Therefore, Hypothesis 3 was not supported. Only phrase patterns supported Hypothesis 3 because one text of GPT-o1 entered the range of distribution of human texts in Fig 1. This conclusion cannot exclude the possibility that more evolved LLMs will be closer to human-written texts in the future.

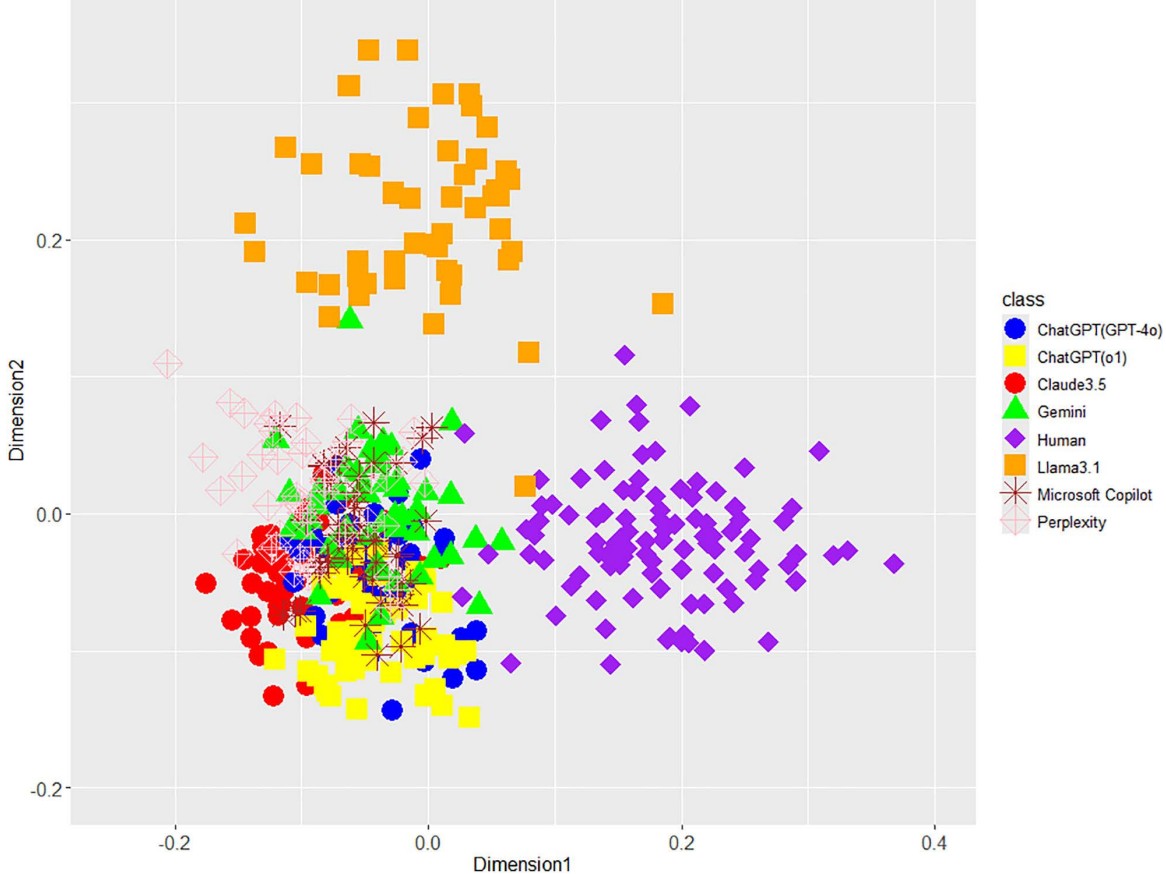

**Fig 2. MDS configuration for POS bigrams in eight classes including each LLM and human.**

## Study 2

### Sample

We utilized the Japanese web survey service, "Questant" provided by Macromill Co., Ltd., and collected responses from the panels on January 24–25, 2025. Though 1,103 Japanese participants responded (667 males, 398 females, 34 participants preferring not to answer, four others, age range 20–73 years (uncertain 19 participants), *Mean* = 48.3, *Median* = 50, *SD* = 11.9), we filtered out participants with inappropriate responses before analysis based on the following criteria for the countermeasure known as "Satisficing [23,24]": (1) Filler items: According to the concept of "Satisficing," a part of survey participants tend not to devote sufficient attentional resources to the survey such as automatically or randomly responding and selecting items without deeper contemplation against each question. Therefore, the present study set the filler text and choices to confirm that the participants deeply selected choices along with instructions, and excluded participants who did not select the choices we instructed from the analyses. (2) Response times: Zhang [24] argued on the relationship between fast responses and "Satisficing." Therefore, participants who answered the survey questionnaire within 3 min were excluded from the analysis. The target participants were 403 (229 males, 164 females, 10 participants preferred not to answer, age range 21–69 years, *Mean* = 47.6, *Median* = 49, *SD* = 11.9).

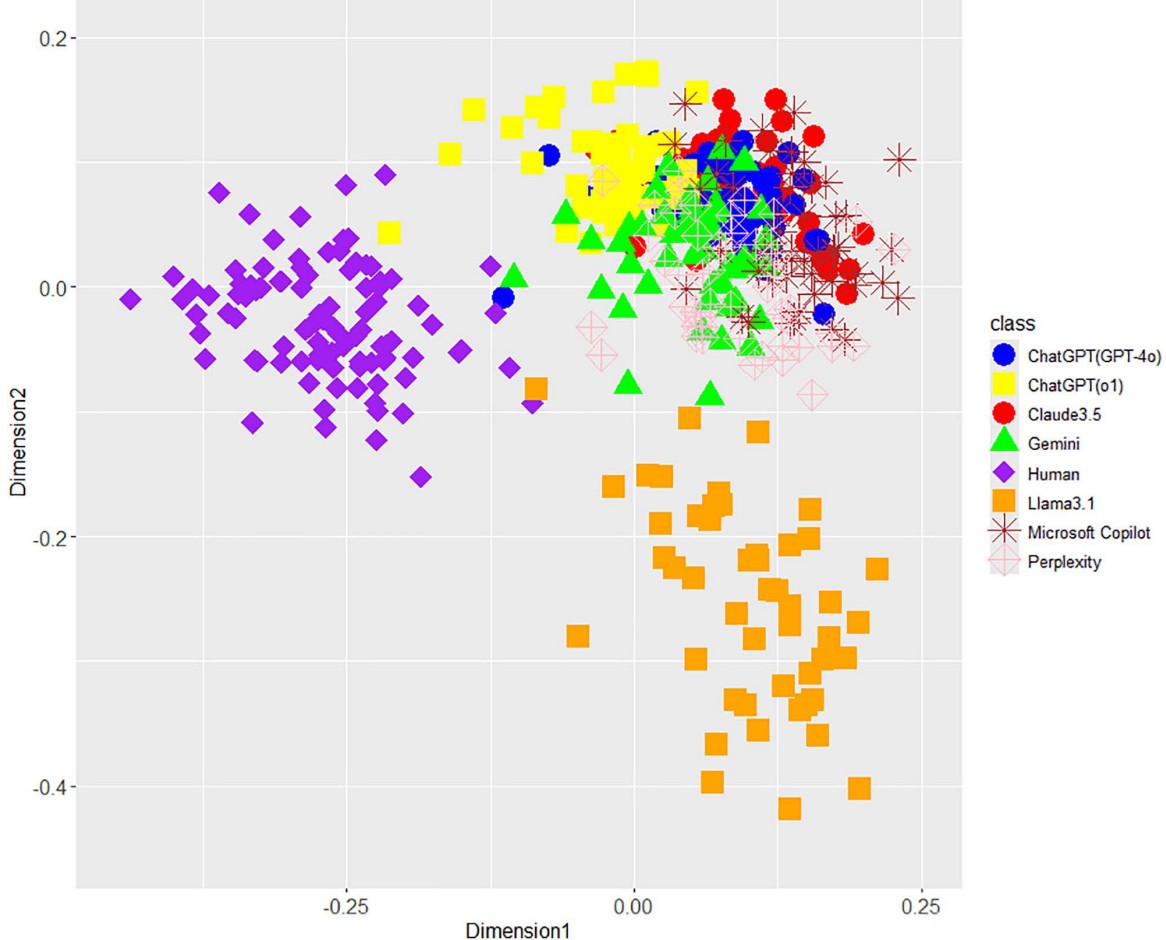

**Fig 3. MDS configuration for phrase patterns in eight classes including each LLM and human.**

## Questionnaires

To assess the accuracy of the survey participants' ability to detect AI and their confidence in their judgments, we conducted a web survey. The questionnaire comprised nine texts (i.e., seven LLM-generated texts, one human-written text, and one filler text) accompanied by two questions: A judgment question ("Do you think this text was written by an AI or by a human?") and a question regarding the confidence of the judgment ("Please rate your level of confidence in this judgment"). Among the texts used in Study 1, eight texts, except for a filler text, were selected based on having the fewest characters to minimize reading burden on participants (i.e., minimum for the effect of "Satisficing [23,24]"). The judgment of the first question was performed by a 7-point Likert scale: 1 = Extremely AI-like, 2 = AI-like, 3 = Somewhat AI-like, 4 = Neither AI nor human for certain (Hereinafter, abbreviated as "Neither"), 5 = Somewhat human-like, 6 = human-like, 7 = Extremely human-like. Additionally, the confidence questions comprised a 6-point Likert scale: 1 = Not confident at all, 2 = Almost no confidence, 3 = Not very confident, 4 = Somewhat confident, 5 = Confident, and 6 = Extremely confident.

After the judgments and confidence estimations, participants were asked to describe the reasons for their decisions in any additional comments. These descriptions were analyzed in terms of several key words associated with their decisions.

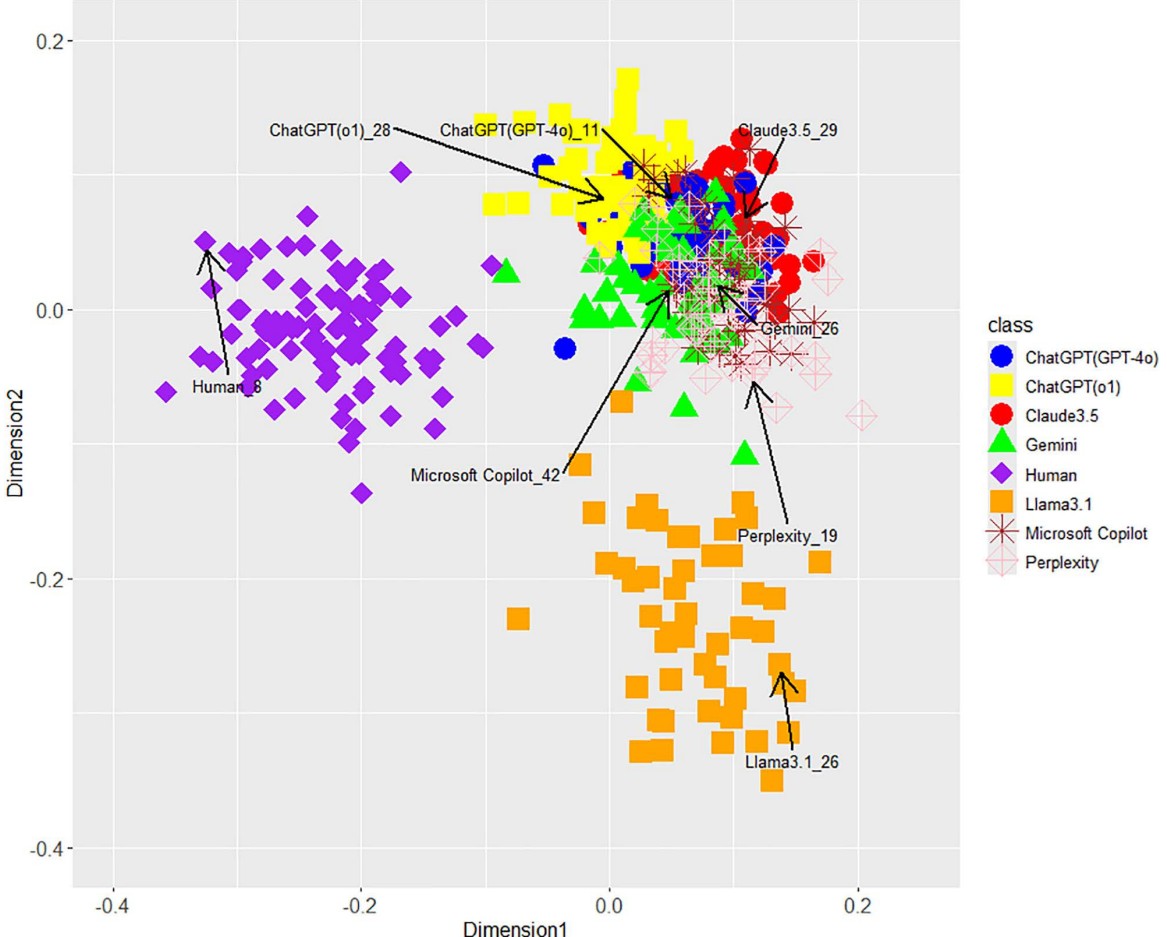

**Fig 4. MDS configuration for integrated features in eight classes including each LLM and human.** * Each text marked by arc and label of LLMs was used as stimulus and presented to the participants in a web survey of Study 2.

## Ethical approval

Ahead of conducting Study2, we applied ethics review and were approved by the Humanities and Social Sciences Research Ethics Committee of MEJIRO University (No. 24人-036). This study ensured the confidentiality of the panel participants by collecting their information anonymously through an online survey. The survey on the web screen provided an explanation that participant was voluntary, and all participants agreed the survey participation and informed consent by clicking a consent button.

## Bayesian logistic regression model with random intercepts for judgments and confidences

To verify Hypotheses 4 and 5 for AI detection in humans, we used Bayesian multinomial logistic regression models with the following linear predictors:

$$\eta_{ir} = ln\left(\frac{P(Y_i = r)}{P(Y_i = LLM)}\right) = \alpha_r + \beta_r \ Text\ type_i + \gamma_r \ Distance\_z_i + u_{j(i),r}$$

$$\alpha_r, \ \beta_r, \ \gamma_r \sim N(0, \ 10^2)$$

$$u_{j(i),r} \sim N(0, \ \sigma_r^2)$$

$$\sigma_r \sim Half-Cauchy(0, \ 5)$$

*i* represents each individual observation (3,224 = 403 participants * 8 judgment estimates per each participant), whereas *j(i)* denotes each participant (1–403). $Y_i$ denotes categorical variables ($Y_i \in \{LLM, \ Human, \ Neither\}$). $\alpha_r$ denotes fixed intercept of category $r \in \{Human, \ Neither\}$ in the set of the reference category $\{LLM\}$ with weakly informative prior distribution [25]. Therefore, $\eta_{ir}$ denotes a log-odds of category $r \in \{Human, \ Neither\}$ against reference category $\{LLM\}$: In the case $r = Human, \eta_{ir} \leq 0$ means $P(Y_i = r) \leq P(Y_i = LLM)$ (i.e., a tendency such as accurate judgment of "LLM"). In contrast, $\eta_{ir} > 0$ indicates $P(Y_i = r) > P(Y_i = LLM)$ (i.e., a tendency to misinterpret "Human"). $\beta_r$ denotes the slope as a fixed effect for explanation variable *Text type*$_{ij}$ with uniform prior distribution similar to $\alpha_r$. In this model, we added the variable of distance as a covariate for controlling them because Fig 1 showed different distances between each LLM and human texts, as presented in Study 2. The values of distance variable were standardized into z-scores Because the self-distance between human-written texts is naturally zero, this study standardized distance as follows: First, we calculated the distances between the target human-written text (presented in Fig 4) and the other 99 human-written texts and computed the mean of these distances (i.e., within-group mean). The original distance value for the human stimulus (e.g., zero) was replaced with the average value. Next, we standardized all distance values based on the mean and standard deviation calculated from the distances between the replaced target human-written text and the other texts (i.e., both LLM-generated and the remaining 99 human-written texts). In addition, this current study employed a within-participants design in which each participant estimated judgments and confidences for all LLMs and human texts. Therefore, we introduced random effect $u_{j(i),r}$ to capture individual differences and adjust for the correlation among repeated measures from the same participant. Moreover, introducing random effects leads to more accurate standard errors and more robust, generalizable fixed effect estimates. The variances of these random effects also have Half-Cauchy prior distribution. In presuming each parameter on Bayes models, we performed Markov Chain Monte Carlo with hyperparameters of chains = 4, iter = 6,000 (the number of samples), warmup = 1,000 (the number of burn-in samples), and thin = 5 (the number of thinning for excluding of autocorrelations) using the *brm* function of the **brms**. We checked the convergence of the posterior distribution by confirming trace plots, autocorrelations, and statistics ($\hat{R} < 1.1$).

As stated above, we modeled the judgment data using participants' response categories ("LLM," "Neither," and "Human") with a baseline-category multinomial logistic regression. The purpose of this approach is to compare the odds of non-reference categories ("Human" and "Neither") with the reference ("LLM"). In contrast, participants' confidence estimates were collected on an ordinal scale ranging from 1 to 6. Therefore, we used a Bayesian ordinal logistic regression model with a cumulative logit link function for confidence estimates, as follows: $C_{ij}$ denotes the score of confidence level (scale 1–6) of participant *j* against text *i*, and $\theta_k$ indicates the parameter of the cut point for determining whether the confidence estimate was *k* or less: In the case of provisional parameters ($\theta_3 = 0.85$, $\alpha = 0$, $\beta = 0.5$, and $u_j = 0.3$), the log odds was $0.05 (= 0.85 - (0 + 0.5 + 0.3))$, and the probability $P(C_{ij} \leq 3)$ based on this the log odds showed $0.512 (= \frac{\exp(0.05)}{1 + \exp(0.05)})$. Under this condition, participants provide a confidence level of 3 or lower for approximately 51.2% of the judgments. In other words, they assigned a confidence level of 4 or higher to approximately 48.8% of cases.

$$ln\left(\frac{P(C_{ij} \leq k)}{P(C_{ij} > k)}\right) = \theta_k - [\alpha + \beta \ \textit{Text type}_i + \gamma \ \textit{Distance\_z}_i + u_j],$$

$$k = 1, \ldots, 5$$

$$\theta_k, \ \alpha, \ \beta, \ \gamma \sim N(0, \ 10^2)$$

$$u_j \sim N(0, \ \sigma_u^2)$$

$$\sigma_u \sim Half-Cauchy(0, \ 5)$$

## Results and discussion in Study 2

Although the details are discussed below, Hypothesis 4 was supported: The accuracy of judgments "LLM or Human" varied across different LLMs. Fig 5 illustrates the violin and box-and-whisker plots of the survey participants' judgments for texts generated by each LLM, as well as those written by humans. Generally, these results indicate the difficulty of the AI detection task because the distributions of participants' judgments were evenly divided into either AI (scale 1–3) or human (scale 5–7) against each LLM-generated and human-written text. Next, we classified the judgment types into three categories ("AI" = 1–3, "Neither AI nor Human" = 4, and "Human" = 5–7) and made a cross tabulation (Table 1) to examine the extent to which the participants' judgments fall into these categories. Table 1 shows that the most accurate judgment for Claude3.5 was 56.8%, that is, 56.8% of the participants correctly assessed the text generated by Claude3.5 as "by LLMs". In contrast, only 31.5% of the participants accurately determined human-written texts as "by Human". The chi-square test for Table 1 showed statistically significant differences between the text type and judgments ($\chi^2$ (14) = 164.9, $p < .01$, Cramer's $V = 0.16$). Moreover, the residual analysis indicated that GPT-4o and Microsoft Copilot were significantly more deceptive (i.e., they succeeded in fooling the participants into thinking that the text was "by human") (each $p < 0.01$). In contrast, according to the residual analysis, the source of the texts generated by Llama3.1 and Perplexity was accurately determined by the participants as "by LLMs", and their judgments are statistically significant (each $p < 0.01$).

Table 2 summarizes the results of the analysis of the Bayesian multinomial logistic regression model for the participants' judgments. The random effects results indicate that the variability of the random intercept for "Neither" is larger than that for "Human": In other words, there is greater intra-individual variation in the tendency to select "Neither". Some participants frequently selected "Neither", but others rarely select "Neither". In case of "Human", if one participant estimated a text as "Human", the others also tend to more consistently select "Human". Regarding the fixed effects for intercept ($\alpha_{Human}$), the log odds (Human/LLM) = −0.10 (i.e., odds ratio (Human/LLM) = exp(−0.10) ≒ 0.90) implies the probabilities that participants were likely to slightly judge "LLM" on the whole on the reference of texts generated by GPT-4o. Moreover, according to $\beta_{Human}$ on six LLMs (except for the reference of GPT-4o) and humans, it can be interpreted that the texts generated by only GPT-o1 (0.87 [0.51, 1.23]) successfully convinced the participants that the texts had been written by a human and that their level of deception was higher than the threshold for change. In particular, GPT-o1 provided the more challenging AI detection task for the participants compared with GPT-4o, as indicated by exp(0.87) ≒ 2.39 (odds ratio). Specifically, the odds ratio 2.39, against 1.00, in favor of Humans, indicates that approximately 70% (= 2.39/ (2.39 + 1)) of judgments are likely to be deceived by GPT-o1. This supports Hypothesis 5: More advanced ChatGPT tends to deceive us. In contrast, the 95% credible interval of $\beta_{Human}$ for the other LLMs included 0 and exhibited no "Human" judgment bias. Interestingly, Llama3.1 texts can be interpreted as having approximately half the odds of being determined as "human-written" compared with GPT-4o texts. According to $\beta_{Neither}$, the texts generated by GPT-o1 are significantly more likely to be judged as "Neither" compared with GPT-4o. In contrast, human-written texts are significantly less likely to be classified as "Neither" compared with GPT-4o.

In summary, the current study demonstrated unreliable accuracy levels of the participants' AI detection in line with previous reports of chance-level human performance [5–6]. In addition, a more advanced LLM (GPT-o1) was successful in deceiving participants that the public comment was written by humans. Therefore, Hypotheses 4 and 5 are supported. Next, we considered the confidence levels associated with the participants' detection accuracies.

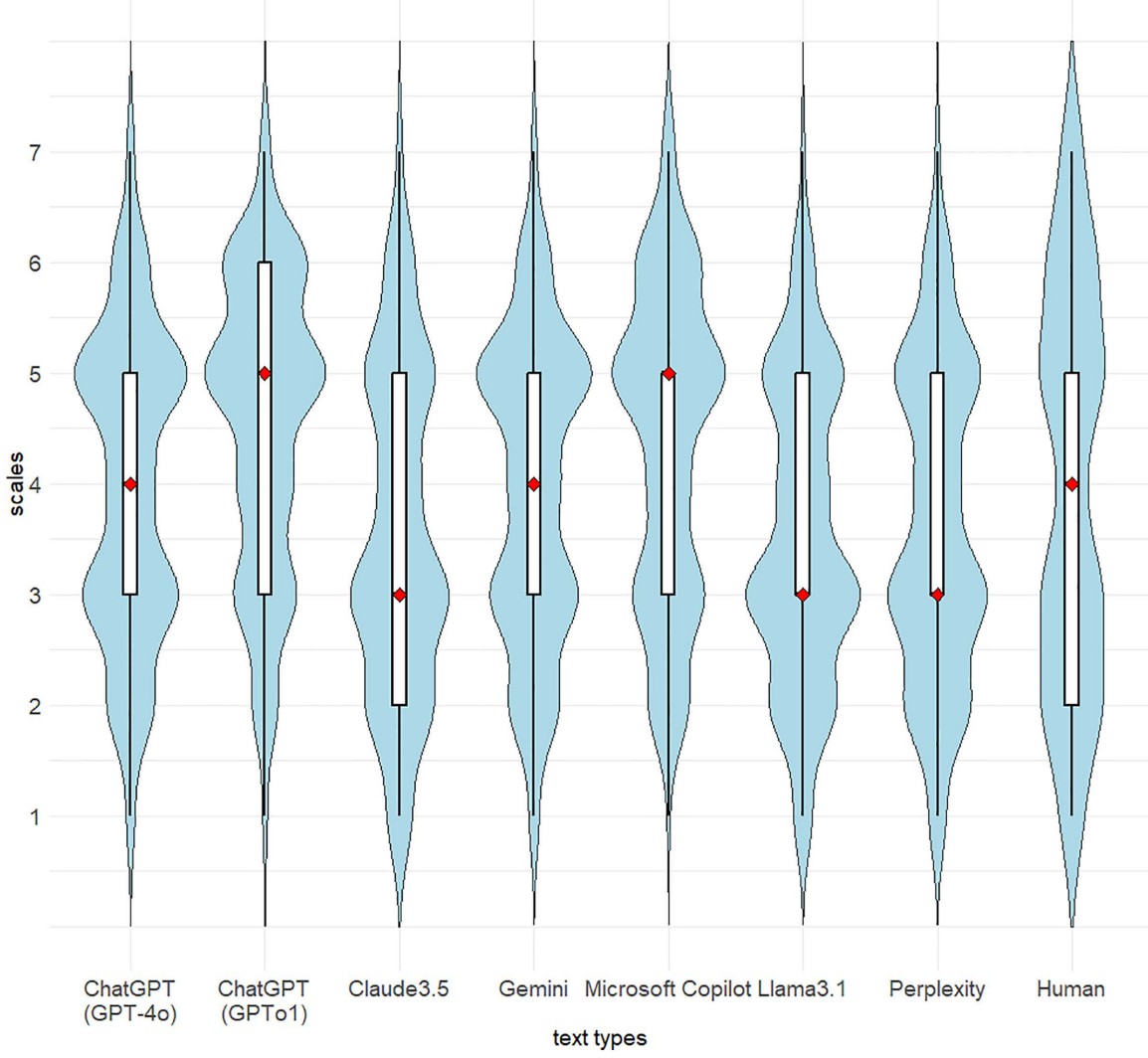

**Fig 5. Violin plots and box-and-whisker plots of judgments of participants (a 7-point Likert scale) against texts corresponding to each LLM and human.** * Each violin plot with light blue shows the shape of the distribution via a kernel density estimate. The red diamond in a box-and-whisker plot represents the median of the distribution.

Fig 6 shows the violin and box-and-whisker plots of the survey participants' confidence in their judgments. The most scales estimated by participants showed "3 = Not very confident" in all LLMs, but only humans' texts enhanced the estimates to "4 = Somewhat confident". Table 3 summarizes the results of the confidence estimates against judgments using the Bayesian ordinal logistic regression model. The reason for absence of $\alpha$ in Table 3 is that the parameters are absorbed into $\theta_k$. From these results, we can interpret the next points for fixed effects: The odds ratio (1.40(=exp(0.34))) for GPT-o1 indicates that the participants had stronger confidences in their own decisions compared to with GPT-4o; Nevertheless, GPT-o1 tended to lead the participants to incorrect judgments as "Human".

Based on additional comments provided after the judgments and confidence estimations by 403 participants, we confirmed subsequent words for reflecting feelings or thoughts for judging as "LLMs": "phraseology ($n$ = 36)", "expression ($n$ = 363)", "unnatural ($n$ = 27)", "usage ($n$ = 25)", "end of words ($n$ = 18)", "conjunction ($n$ = 15)", "discomfort ($n$ = 13)", and

**Table 1. Cross tabulation for text types (generated by LLMs and written by human) and judgments of survey participants.**

| Source Texts | Judgments of participants | | |
|---|---|---|---|
| | LLMs | Neither LLMs nor Human | Human |
| GPT-4o | 172 (42.7%) | 52 (12.9%) | 179 (44.4%) |
| GPT-o1 | 103 (25.6%)** | 61 (15.1%) | 239 (59.3%)** |
| Claude3.5 | 229 (56.8%) | 47 (11.7%) | 127 (31.5%) |
| Gemini | 169 (41.9%) | 57 (14.1%) | 177 (43.9%) |
| Microsoft Copilot | 134 (33.3%)** | 47 (11.7%) | 222 (55.1%)** |
| Llama3.1 | 225 (55.8%)** | 56 (13.9%) | 122 (30.3%)** |
| Perplexity | 213 (52.9%)** | 50 (12.4%) | 140 (34.7%)** |
| Human | 193 (47.9%) | 28 (6.9%)** | 182 (45.2%) |

Residual analysis: $^{*}p < .05$, $^{**}p < .01$.

**Table 2. Results of analysis for judgments of participants using Bayesian multinomial logistic regression model with random intercepts.**

| Effect | Parameter | Estimate | 95% Credible Interval | Standard Error | $\hat{R}$ |
|---|---|---|---|---|---|
| Random Effect | $\sigma^2_{Human}$ | 0.63 | [0.50, 0.77] | 0.07 | 1.00 |
| | $\sigma^2_{Neither}$ | 1.53 | [1.30, 1.80] | 0.13 | 1.00 |
| Fixed Effect | $\alpha_{Human}$ | −0.10 | [-2.94, 2.67] | 1.44 | 1.00 |
| | $\alpha_{Neither}$ | −1.92 | [-4.78, 0.89] | 1.46 | 1.00 |
| | $\beta_{Human}(GPT-o1)$ | 0.87 | [0.51, 1.23] | 0.18 | 1.00 |
| | $\beta_{Human}(Claude3.5)$ | −0.47 | [-5.60, 4.81] | 2.67 | 1.00 |
| | $\beta_{Human}(Gemini)$ | 0.20 | [-4.41, 4.95] | 2.42 | 1.00 |
| | $\beta_{Human}(Llama3.1)$ | −0.09 | [-14.82, 15.30] | 7.80 | 1.00 |
| | $\beta_{Human}(Microsoft\ Copilot)$ | 0.65 | [-2.79, 4.18] | 1.80 | 1.00 |
| | $\beta_{Human}(Perplexity)$ | −0.38 | [-3.34, 2.66] | 1.55 | 1.00 |
| | $\beta_{Human}(Human)$ | −0.26 | [-4.35, 3.73] | 2.10 | 1.00 |
| | $\gamma_{Human}$ | −0.29 | [-7.43, 6.56] | 3.63 | 1.00 |
| | $\beta_{Neither}(GPT-o1)$ | 0.74 | [0.22, 1.25] | 0.26 | 1.00 |
| | $\beta_{Neither}(Claude3.5)$ | −0.39 | [-5.62, 4.81] | 2.70 | 1.00 |
| | $\beta_{Neither}(Gemini)$ | 0.17 | [-4.53, 4.91] | 2.44 | 1.00 |
| | $\beta_{Neither}(Llama3.1)$ | −0.08 | [-15.20, 15.23] | 7.87 | 1.00 |
| | $\beta_{Neither}(Microsoft\ Copilot)$ | 0.14 | [-3.40, 3.74] | 1.82 | 1.00 |
| | $\beta_{Neither}(Perplexity)$ | −0.25 | [-3.29, 2.77] | 1.57 | 1.00 |
| | $\beta_{Neither}(Human)$ | −0.94 | [-5.11, 3.24] | 2.15 | 1.00 |
| | $\gamma_{Neither}$ | −0.04 | [-7.13, 7.06] | 3.67 | 1.00 |

"punctuation mark ($n$ = 12)". According to these words, people appear to make their own judgments impressionistically based on linguistically lower-level or surface information.

Although the aforementioned results were obtained, Study 2 was limited because we presented only one public comment for each LLM. Therefore, the content of the test texts is not controlled in an equal manner, which would allow for unbiased judgments; nevertheless, Study 2 attempted to control distances for each stylometric feature between humans

and LLMs. It is assumed that participants' impressions of public comments are somewhat different despite the same LLM-generated texts because of the impressions produced by different content. Therefore, it is necessary to continue similar surveys by presenting participants with different LLMs texts and by controlling the content.

## General discussion

In Study 1, we attempted to distinguish the public comments generated by seven LLMs from those written by humans by focusing on three Japanese stylometric features (function word unigrams, POS bigrams, and phrase patterns). The MDS and RF results revealed the potential for complete AI detection and supported Hypothesis 1. This result coincided with those of previous studies [10,11]. Successful AI detection can be attributed to the adoption of effective methods in Japanese authorship analysis: Stylometric features (function word unigrams, POS bigrams, and phrase patterns), distance

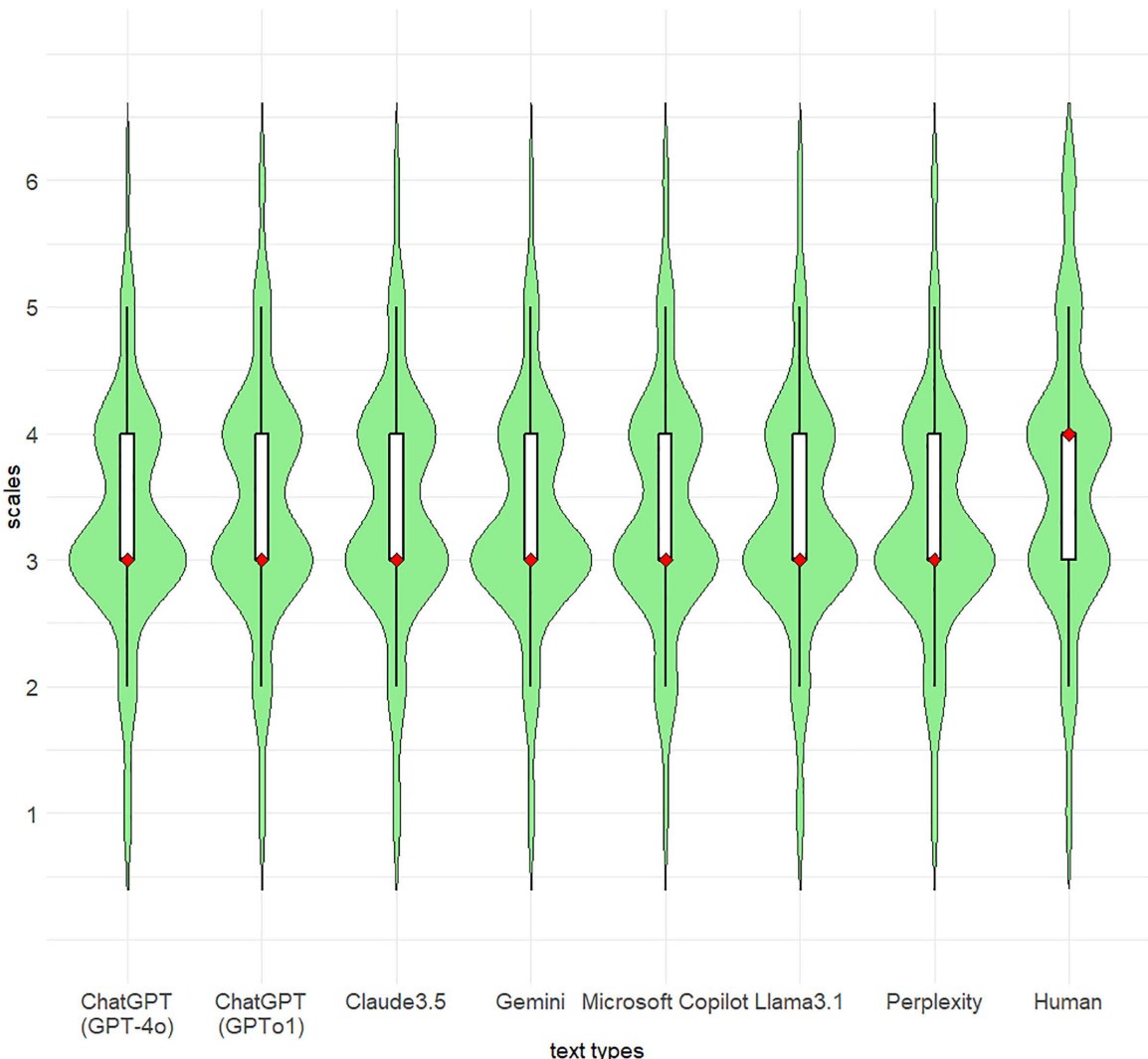

**Fig 6. Violin plots and box-and-whisker plots of confidence of participants (a 6-point Likert scale) against texts corresponding to each LLM and human.** * Each violin plot with light green shows the shape of the distribution via a kernel density estimate. The red diamond in a box-and-whisker plot represents the median of the distribution.

metrics ($d_{SJSD}$), and analytical methods (MDS and RF). In addition, MDS clarified the similarities of each LLM from the perspective of stylometric features, but only Llama3.1 had distinct stylometric features among the other LLMs, and Hypothesis 2 was partially supported. Why was only Llama3.1 apart from the other LLMs in the MDS dimensions? To the best of our knowledge, Llama was trained mainly on CommonCrawl [26]. Although not all LLMs have been made public, no difference may be present in terms of source data. Additionally, recent LLMs use common transformers [16]. However, the number of parameters of Llama, ranging from 7B to 65B parameters [26], is relatively small compared with those of other LLMs (e.g., ChatGPT and Gemini). Therefore, it is possible hypothesis that the number of parameters is related to distinct stylometric features. Therefore, a more detailed investigation is required. Concerning Hypothesis 3, no significant difference was observed between the advanced LLMs (GPT-o1) and older LLMs (GPT-4o) in terms of stylometric features. This finding is consistent with those of previous studies [10]. Next, Study 2 performed a web survey to support Hypotheses 4 and 5. These findings matched to those of previous studies [5–8] asserted that humans' AI detection abilities are limited. Additionally, to make matters worse, we discovered the possibility that more advanced LLMs (e.g., GPT-o1) might lead people to mistakenly believe "by Human", as demonstrated by Clark et al. [6], and make individuals have more confidence in their wrong judgments.

By verifying Hypotheses 1–5, the conclusions suggest that if more advanced LLMs are developed, humans will be more easily deceived, because laypeople tend to make their judgments impressionistically, primarily based on first-hand surface textual information. Stylometric analysis, which enables us to extract linguistically higher-level or deeper information, including function word unigrams, POS bigrams, and phrase patterns, can relatively easily expose the source of a text, either LLM-generated or human-authored, because these features may not be more similar to those of humans. The unique properties of texts to consider and the resultant cognitive processes leading to judgment are distinct between stylometric analysts and humans. It is easy to foresee that this can lead to various problems. For instance, government officers may be unaware of AI-generated public comments, university professors may overlook LLM-generated reports from students, and consumers may be misled by AI-generated fake reviews. This problem will intensify with the continued development of LLMs. Furthermore, from the perspective of psychology, individuals tend to have various cognitive biases, such as authority and confirmation biases [27], which are influenced by the contextual, environmental, and situational factors surrounding reading texts. Consequently, judgments under cognitive biases may deteriorate depending on the

**Table 3. Results of analysis for confidences of participants using Bayesian ordinal logistic regression model with random intercepts.**

| Effect | Parameter | Estimate | 95% Credible Interval | Standard Error | $\hat{R}$ |
|---|---|---|---|---|---|
| Random Effect | $\sigma_u^2$ | 2.04 | [1.86, 2.23] | 0.09 | 1.00 |
| Fixed Effect | $\theta_1$ | −5.43 | [-8.36, -2.50] | 1.50 | 1.00 |
| | $\theta_2$ | −2.98 | [-5.89, -0.06] | 1.49 | 1.00 |
| | $\theta_3$ | 0.89 | [-2.00, 3.80] | 1.49 | 1.00 |
| | $\theta_4$ | 3.62 | [0.76, 6.51] | 1.49 | 1.00 |
| | $\theta_5$ | 5.46 | [2.55, 8.42] | 1.49 | 1.00 |
| | $\beta(GPT-o1)$ | 0.34 | [0.03, 0.67] | 0.16 | 1.00 |
| | $\beta(Claude3.5)$ | 0.36 | [-4.91, 5.71] | 2.74 | 1.00 |
| | $\beta(Gemini)$ | 0.13 | [-4.65, 5.04] | 2.48 | 1.00 |
| | $\beta(Llama3.1)$ | 0.21 | [-15.36, 15.99] | 8.01 | 1.00 |
| | $\beta(Microsoft\ Copilot)$ | 0.29 | [-3.29, 3.95] | 1.85 | 1.00 |
| | $\beta(Perplexity)$ | −0.06 | [-3.14, 3.10] | 1.59 | 1.00 |
| | $\beta(Human)$ | 0.93 | [-3.33, 5.11] | 2.16 | 1.00 |
| | $\gamma$ | −0.08 | [-7.40, 7.16] | 3.73 | 1.00 |

situation. However, to the best of our knowledge, no AI detection tools available in the market for specializing in Japanese texts have shown high accuracy, including recall, precision, and F1. Therefore, the development of highly reliable AI detection tools is urgently required.

We obtained the above results, but the next main limitations should be noted. (1) In both studies, we analyzed only Japanese public comments. However, writing style in Japanese is likely influenced by sentence formality (formal vs. informal), constraints imposed by format (e.g., academic papers), and genre (e.g., creative writing, essays, social media, email). Therefore, the above results may limit generalizability, it is necessary to target a wider range of media to confirm this study's conclusions. (2) In Study 2, the experimenters presented only one text in each LLM to participants. Texts generated by same LLMs were different in terms of content, wording, and phrasing in addition to writing style as Figs 1–4 showed distributions. We attempted to control only writing-style biases within same LLMs by including distance differences into the Bayesian model, but we did not control the other factors. Future studies should present multiple texts per LLM and evaluate content effects to improve generalizability. (3) This study limited only texts generated via zero-shot prompting. As stated in the Introduction, prior work [11] reports that texts generated by one-shot prompting are more challenging to analyze compared with ones via zero-shot prompting. In future work, it will be necessary to examine the distributions of texts generated by one- or few-shot prompting.

## Future studies

This study clarified the effectiveness of stylometric analysis in detecting AI-generated texts and identified the difficulties faced by laypersons when determining AI-generated texts using visual impressions. At the end of our study, several new questions and directions for future research emerged. For instance, why could we distinguish texts produced by both LLMs and humans from the perspective of stylometric analysis, despite each LLM being pretrained on the basis of human-written texts? Do other LLMs have unique stylometric features such as Llama3.1? Factors such as sex, age, job, and frequency of generative AI usage explain individual differences in AI detection. Can college professors who are accustomed to reading student reports easily distinguish the source of authorship (AI or human) from laypersons? It is necessary to examine the underlying causes of these phenomena and elucidate fundamental questions from multiple perspectives.

## Supporting information

**S1 Data. Study1_FunctionWords.**
(CSV)

**S2 Data. Study1_POS_bigram.**
(CSV)

**S3 Data. Study1_PhrasePatterns.**
(CSV)

**S4 Data. Study2_judgment.**
(CSV)

## Author contributions

**Conceptualization:** Wataru Zaitsu, Mingzhe Jin, Shunichi Ishihara, Satoru Tsuge, Mitsuyuki Inaba.

**Data curation:** Wataru Zaitsu.

**Formal analysis:** Wataru Zaitsu, Mingzhe Jin.

**Funding acquisition:** Wataru Zaitsu, Satoru Tsuge, Mitsuyuki Inaba.

**Investigation:** Wataru Zaitsu.

**Methodology:** Wataru Zaitsu, Shunichi Ishihara, Satoru Tsuge.

**Project administration:** Wataru Zaitsu, Mingzhe Jin.

**Supervision:** Mingzhe Jin, Shunichi Ishihara.

**Visualization:** Wataru Zaitsu.

**Writing – original draft:** Wataru Zaitsu, Shunichi Ishihara.

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
