## [Decision Letter · Decision Letter 0]

4 Sep 2025

Dear Dr.  Zaitsu,

Thank you for submitting your manuscript to PLOS ONE. After careful consideration, we feel that it has merit but does not fully meet PLOS ONE’s publication criteria as it currently stands. Therefore, we invite you to submit a revised version of the manuscript that addresses the points raised during the review process.

We look forward to receiving your revised manuscript.

Kind regards,

Emine Ozdemir Kacer

Academic Editor

PLOS ONE

Journal Requirements:

Reviewer's Responses to Questions

**Comments to the Author**

1. Is the manuscript technically sound, and do the data support the conclusions?

Reviewer #1: Partly

Reviewer #2: Yes

Reviewer #3: Yes

Reviewer #4: No

2. Has the statistical analysis been performed appropriately and rigorously?

Reviewer #1: Yes

Reviewer #2: Yes

Reviewer #3: Yes

Reviewer #4: No

3. Have the authors made all data underlying the findings in their manuscript fully available?

Reviewer #1: No

Reviewer #2: Yes

Reviewer #3: Yes

Reviewer #4: Yes

4. Is the manuscript presented in an intelligible fashion and written in standard English?

Reviewer #1: No

Reviewer #2: Yes

Reviewer #3: Yes

Reviewer #4: Yes

Reviewer #1: Summary

This manuscript presents two complementary studies on distinguishing AI-generated from human-written Japanese text. In Study 1, the authors compare 350 texts from seven LLMs (ChatGPT variants, Claude, Gemini, Copilot, Llama3.1, Perplexity) against 100 human-written public comments using Japanese stylometric features (function-word unigrams, POS bigrams, and phrase patterns). Multidimensional scaling (MDS) and a random-forest classifier (with leave-one-out cross-validation) showed that stylometry can almost perfectly separate AI vs. human text (RF accuracy ~99.8%). In Study 2, 403 Japanese participants judged a set of 9 example texts (eight LLM-generated, one human) as “AI” or “Human” and rated their confidence. Participants performed poorly (often at chance) and were especially fooled by more advanced models (ChatGPT(o1)), aligning with prior findings that untrained humans struggle to detect AI-generated text. The topic is timely given the rapid evolution of LLMs, and the use of stylometry for authorship analysis is well-motivated.

Major Comments

Scope and Generalizability: The corpus in both studies is limited to public comments on government documents. While this domain allows controlled comparisons, it is quite specific. The authors should explicitly acknowledge that stylometric patterns may differ in other genres (e.g. creative writing, social media) or languages. Clarify in the Discussion that findings may not generalize beyond Japanese public-comment texts. It would strengthen the paper to mention whether the authors plan to test other text types in future work.

Classification Methodology: The authors use a random forest (RF) with leave-one-out cross-validation (LOOCV) to classify texts (Study 1). This approach is reasonable to avoid overfitting, but some details are missing. The paper should specify RF parameters (number of trees, depth, etc.) and whether class imbalance was handled (350 AI vs 100 human texts). The reported accuracy (99.8%) is extremely high. To validate this, the authors could report a confusion matrix or feature importance to ensure no trivial features dominate. For example, length differences (human texts avg 661 chars vs LLM texts up to 1137 chars) might partly drive the result; clarifying that features were normalized (which they were, by percentage) is good. Overall, the methodology seems sound, but more transparency about the RF and CV would help readers assess reproducibility.

Study 2 Design Limitation: Study 2’s design is inherently constrained by using only one example per LLM. The authors themselves note: “Study 2 was limited because we presented only one public comment for each LLM”. This is a serious limitation. With a single text per model, any idiosyncratic content or phrasing could bias participants’ judgments. The manuscript should emphasize how this limitation affects the interpretation: participants might have been reacting to the specific text content rather than general style. A stronger discussion of this point is needed. For future work, multiple texts per LLM (and human) should be used to average out content effects. The authors should make clear that Study 2 primarily illustrates feasibility rather than providing definitive, generalizable detection rates.

Connection to Prior Work: The Introduction and Discussion cite relevant studies, but the paper could better contextualize the human-evaluation results. For example, Clark et al. (2021) found that untrained evaluators distinguish GPT-3 from human text only at random chance

aclanthology.org.

Similarly, Köbis & Mossink (2020) report that participants failed to reliably detect GPT-2–generated poetry

arxiv.org.

The current findings (participants’ confusion, low accuracy) align well with these, but the paper should explicitly mention such parallels. Citing these works in the Discussion would strengthen the claim that “humans’ AI detection abilities are limited.” The authors do cite Clark et al. in passing, but an explicit linkage to these results and perhaps a brief summary (e.g. “in line with previous reports of chance-level human performance

aclanthology.org

arxiv.org

”) would help readers understand the broader context.

Interpretation of Stylometric Differences: The finding that Llama3.1 texts occupy a different stylistic space (Fig 1–4) is interesting. The authors speculate it might be due to Llama’s smaller parameter count. This is plausible, but is speculative. The Discussion should present this more cautiously (e.g. “one possibility is”). Other factors (different training data, tokenization, or style by design) could also play a role. If possible, the authors could consider a simple follow-up analysis (e.g. train RF to differentiate Llama vs other LLMs) or at least frame the parameter-size explanation as a hypothesis.

Presentation of Results: Some results would benefit from clearer explanation. For example, the logistic regression on confidence (Table 3) is technically sound, but the text could better highlight the main takeaway (ChatGPT(o1) significantly increased odds of human-like judgments and higher confidence). Also, the labeling in Table 1 (participants’ judgments) is a bit dense; the authors might spell out key points in the text. More generally, ensure every table and figure is referenced and explained clearly.

Minor Comments

Language and Typos: The manuscript contains numerous small language issues. These should be carefully corrected. Examples include: "detection of disabilities by humans" (L691-692) – likely meant “detection by humans” or “detection difficulties for humans”. In the Abstract, “overly AI detection abilities were limited” should read “overall AI detection ability was limited.” The term “judgement confidents” appears (L1955-1958) – it should be “confidence.” The acronym mix-up “LMM-generated” (L864) should be “LLM-generated.” Please proofread for such errors. It may help to have a native English speaker review the phrasing.

Clarity of Terms: The manuscript uses “ChatGPT (GPT-4o and o1)” and later simply “ChatGPT(o1)” vs “ChatGPT(GPT-4o)”. This notation is confusing. It would be clearer to fully name these variants once (e.g. “ChatGPT (based on GPT-4.0)” vs “ChatGPT (GPT-4o1)”) and then use consistent shorthand. Similarly, the term “LLM (Large Language Model)” should be defined at first use in the Introduction.

Figure Captions: Ensure all figures have descriptive captions and legends. The text refers to “Fig 1–4” but readers should not have to guess axes or symbols. Since the file we received lacked figure content, I cannot check fully, but reviewers should verify that figure captions are complete in the submitted version.

Data Availability: The manuscript’s Data Availability statement is incomplete. PLOS ONE requires that all data (the Japanese text corpus, survey responses, etc.) be made available. The authors should state where data and code can be accessed (e.g. a public repository) or justify any restrictions. If not already included, please add a full Data Availability statement.

Ethics Statement: Study 2 involved human participants. It is good that ethics approval from Mejiro University is mentioned (Ethics Committee No.24人-036). Ensure this is clearly stated in the Methods (it appears only in the submission interface and references). It might also be helpful to mention that participants gave informed consent.

Reference to Self-Published Work: The authors cite their prior PLOS ONE studies [10–11]. While building on those is fine, the manuscript should emphasize the new contributions here (multiple new LLMs, human survey). Avoid reusing large verbatim text from those works. (I did not detect obvious overlap, but it is worth verifying against [44]).

Recommendation

This work addresses an important and timely issue with a generally sound approach and interesting findings. The combination of stylometric classification and human judgment experiments is valuable. However, the manuscript needs minor revisions before it is suitable for publication. Key improvements include: clarifying methodological details (e.g. RF parameters, cross-validation), acknowledging the limitations of the survey design more strongly, and correcting language/typos. Addressing the above points will considerably strengthen the paper.

Reviewer #2: The manuscript, “Stylometry can reveal artificial intelligence authorship, but humans struggle: A comparison of human and seven large language models in Japanese”, presents a technically sound and well-structured study. It addresses an important and timely issue by comparing machine-based stylometric detection with human judgments in the Japanese language context. The methodology is rigorous, the analyses are appropriate, and the data support the conclusions. Ethical approval and data availability are also adequately addressed.

There are, however, several areas where the manuscript could be strengthened:

1- In Study 2 design ,Only one text per LLM was used in the human judgment task. This limits the generalizability of the findings, as responses may partly reflect topic effects rather than model stylistics. While the authors acknowledge this, it should be emphasized more strongly, with clearer discussion of how future studies could incorporate multiple texts per model.

2- Although generally intelligible, some phrases are awkward (as “overly AI detection abilities were limited”). A professional English edit would improve readability and ensure the results and interpretations are conveyed with precision.

3- The MDS and violin plots are central to the findings but may not be immediately interpretable for all readers. Figure legends should be expanded to explain more clearly what the axes, scales, and distributions represent.

4- The finding that ChatGPT-o1 “misleads humans more effectively” is supported, but the discussion should avoid overgeneralization. Alternative explanations, including topic familiarity, surface-level linguistic cues, or cognitive biases, should be considered more explicitly.

Reviewer #3: The design of this complex study and the technical application of several complex statistical analyses are well done and documented clearly in the manuscript. The explanation of the data in support of the five clearly delineated hypotheses is expertly and concisely done. The data source is appropriately identified, and ethical standards review is documented. Illustrated formulas support documentation of statistical analysis, and tables are relevant. The manuscript is well written and organized for the complexity of the study.

Reviewer #4: Comments to authors:

1. The authors restrict their analysis to three features (phrase patterns, part-of-speech bigrams, and unigrams of function words). While these are established, the exclusion of higher-level syntactic or semantic features (e.g., dependency relations, embeddings-based features) is a limitation. The paper should justify why only these features were used and discuss whether the results might change with richer stylometric representations.

2. The multinomial logistic regression model includes distance as a covariate, but the authors do not clearly explain how distances were computed or standardized across features. More methodological clarity is needed.

3. The reliance on metric MDS is justified in terms of interpretability, but other dimensionality reduction techniques (e.g., t-SNE, PCA) might capture non-linear relationships more effectively. A comparative analysis, or at least a sensitivity check, would strengthen the claim that MDS is the best choice.

4. The Random Forest classifier achieved near-perfect accuracy (99.8%), which raises concerns of overfitting. Was cross-validation (other than LOOCV) performed? Were hyperparameters tuned? Were the results stable across different random seeds? The authors should provide more transparency on classifier robustness.

5. The ‘human-written’ texts come only from the Japanese government's public comments. This corpus is domain-specific, formal, and possibly homogeneous. The findings may not generalize to other text genres (e.g., creative writing, casual conversation). The authors should either include a second human corpus or clearly acknowledge this limitation.

6. All LLM outputs were generated with zero-shot prompts. However, the prompting strategy strongly influences output style. Without testing few-shot or instruction-tuned prompts, the conclusions about stylometric detectability may not generalize. At a minimum, this limitation should be discussed.

7. The authors should write a limitations section.

**Do you want your identity to be public for this peer review?** For information about this choice, including consent withdrawal, please see our Privacy Policy

Reviewer #1: No

Reviewer #2: No

Reviewer #3: No

Reviewer #4: No

---

## [Author Response · Author response to Decision Letter 1]

7 Sep 2025

Answer and Response to Reviewer 1 comments:

Thank you for your constructive review.

I revised several points along with your comments.

Please confirm revised paper.

Thank you.

Major Comments

1. Scope and Generalizability

We added limitations in final of general discussion along with your comment as follows:

“We obtained the above results, but the next main limitations should be noted. (1) In both studies, we analyzed only Japanese public comments. However, writing style in Japanese is likely influenced by sentence formality (formal vs. informal), constraints imposed by format (e.g., academic papers), and genre (e.g., creative writing, essays, social media, email). Therefore, the above results may limit generalizability, it is necessary to target a wider range of media to confirm this study’s conclusions.”

2. Classification Methodology:

Concerned this points, we think it is enough to demonstrate clear different distribution between LLMs and Human by using MDS. So RF classifier is just supplementary analysis because it is natural that they can be neatly classified.

But for reference we conducted sensitivity analysis as follows:

“In addition, sensitivity analyses, tuned across each hyperparameter (ntree, mtry, maxnodes, and nodesize) together with down-sampling to balance classes from AI=350/Human=100 to AI=100/Human=100, also showed stable performances.”

3.Study 2 Design Limitation:

We added limitations in final of general discussion along with your comment as follows:

“(2) In Study 2, the experimenters presented only one text in each LLM to participants. Texts generated by same LLMs were different in terms of content, wording, and phrasing in addition to writing style as Fig 1–4 showed distributions. We attempted to control only writing-style biases within same LLMs by including distance differences into the Bayesian model, but we did not control the other factors. Future studies should present multiple texts per LLM and evaluate content effects to improve generalizability.”

4. Connection to Prior Work

We revised paragraph before Table2.

“In summary, the current study demonstrated unreliable accuracy levels of the participants’ AI detection in line with previous reports of chance-level human performance [5-6].”

5. Interpretation of Stylometric Differences:

We revised this point by only adding one word “hypothesis”.

“(l.458) Therefore, it is possible hypothesis that the number of parameters is related to distinct stylometric features.”

6. Presentation of Results:

In this point, we have already intended to write in each key point (e.g., The odds rate (1.40(=exp(0.34)) for GPT-o1 indicates that the participants had stronger confidences in their own decisions compared to with GPT-4o). OK?

Minor Comments

Language and Typos:

We revised sentences along with your comments.

By the way, we had already a native person check the manuscript.

Clarity of Terms:

We revised as “GPT-4o” and “GPT-o1”.

Figure Captions:

We added captions as follows:

Fig 1. MDS configuration for function word unigrams in eight classes including each LLM and human.

* The distances (e.g., symmetric Jensen–Shannon distances) between each object on the two dimensions indicate relative similarity of concerned stylometric features.

Data Availability:

We will reposit our dataset on PLOS ONE site and intend to write that down in the input form.

Ethics Statement:

We moved “ethics statements and informed consent” into Methods along with your comment.

Reference to Self-Published Work:

Along with your comment, we added the following sentence in introduction.

“In belief, this study extended these previous studies by evaluating new text (e.g., public comments) using multiple LLMs and by examining a human-judgment as stated next.”

Answer and Response to Reviewer 2 comments:

Thank you for your constructive review.

I revised several points along with your comments.

Please confirm revised paper.

Thank you.

1. Study 2 Design Limitation:

We added limitations in final of general discussion along with your comment as follows:

“(2) In Study 2, the experimenters presented only one text in each LLM to participants. Texts generated by same LLMs were different in terms of content, wording, and phrasing in addition to writing style as Fig 1–4 showed distributions. We attempted to control only writing-style biases within same LLMs by including distance differences into the Bayesian model, but we did not control the other factors. Future studies should present multiple texts per LLM and evaluate content effects to improve generalizability.”

2. Language and Typos:

We revised sentences along with your comments.

By the way, we had already a native person check the manuscript.

3. MDS and violin plots

We added captions of MDS as follows:

Fig 1. MDS configuration for function word unigrams in eight classes including each LLM and human.

* The distances (e.g., symmetric Jensen–Shannon distances) between each object on the two dimensions indicate relative similarity of concerned stylometric features.

But caption of violin plots have been written as follows.

“Fig 5. Violin plots and box-and-whisker plots of judgments of participants (a 7-point Likert scale) against texts corresponding to each LLM and human

* Each violin plot with light blue shows the shape of the distribution via a kernel density estimate. The red diamond in a box-and-whisker plot represents the median of the distribution.”

4. The finding for ChatGPT-o1

We revised along with your comment as follows in abstract.

“Moreover, in our materials, more advanced ChatGPT(o1), plausibly reflecting relatively greater fluency and polish, tends to mislead the participants to believe “human-written” texts compared with ChatGPT(GPT-4o) and improves their confidence for their own judgments.”

Answer and Response to Reviewer 3 comments:

Thank you for your constructive review.

I revised several points along with other reviewers’ comments.

Please confirm revised paper.

Thank you.

Answer and Response to Reviewer 4 comments:

Thank you for your constructive review.

I revised several points along with your comments.

Please confirm revised paper.

Thank you.

1. Restrict of writing styles

We focused on only three features (phrase patterns, POS bigrams, and function words unigrams) because we expected them to provide sufficient discriminatory power. As we expected, our results confirmed this. Exploring higher-level features is valuable but was outside the scope of the present study and is not required for our current conclusions.

2. Standardized distances

We have already written this method of standardized distances in Supporting information or footnote. But we made some modifications as follows.

“*1 Because the self-distance between human-written texts is naturally zero, this study standardized distance as follows: First, we calculated the distances between the target human-written text (presented in Fig 4) and the other 99 human-written texts and computed the mean of these distances (i.e., within-group mean). The original distance value for the human stimulus (e.g., zero) was replaced with the average value. Next, we standardized all distance values based on the mean and standard deviation calculated from the distances between the replaced target human-written text and the other texts (i.e., both LLM-generated and the remaining 99 human-written texts).”

3. The other methods

As explained in the previous article, we did not use t-SNE because of reproducibility with random initialization. Moreover, MDS is considered as valid method compared with other statistical method (e.g., PCA) for Japanese authorship verification. Furthermore, we think that given that our metric-MDS map already cleanly separates the classes and it is not necessary to use further alternative methods.

4. Random Forest:

Concerned this point, we think it is enough to demonstrate clear different distribution between LLMs and Human by using MDS. So RF classifier is just supplementary analysis because it is natural that they can be neatly classified.

But for reference we conducted sensitivity analysis as follows:

“In addition, sensitivity analyses, tuned across each hyperparameter (ntree, mtry, maxnodes, and nodesize) together with down-sampling to balance classes from AI=350/Human=100 to AI=100/Human=100, also showed stable performances.”

5. Text genre and Generalizability

We added limitations in final of general discussion along with your comment as follows:

“We obtained the above results, but the next main limitations should be noted. (1) In both studies, we analyzed only Japanese public comments. However, writing style in Japanese is likely influenced by sentence formality (formal vs. informal), constraints imposed by format (e.g., academic papers), and genre (e.g., creative writing, essays, social media, email). Therefore, the above results may limit generalizability, it is necessary to target a wider range of media to confirm this study’s conclusions.”

6. Zero-shot prompting:

We added limitations in final of general discussion along with your comment as follows:

“(3) This study limited only texts generated via zero-shot prompting. As stated in the Introduction, prior work [11] reports that texts generated by one-shot prompting are more challenging to analyze compared with ones via zero-shot prompting. In future work, it will be necessary to examine the distributions of texts generated by one- or few-shot prompting.”

---

## [Decision Letter · Decision Letter 1]

10 Oct 2025

Stylometry can reveal artificial intelligence authorship, but humans struggle: A comparison of human and seven large language models in Japanese

PONE-D-25-18282R1

Dear Dr. Zaitsu,

We’re pleased to inform you that your manuscript has been judged scientifically suitable for publication and will be formally accepted for publication once it meets all outstanding technical requirements.

Kind regards,

Emine Ozdemir Kacer

Academic Editor

PLOS ONE

Additional Editor Comments (optional):

Reviewers' comments:

Reviewer's Responses to Questions

**Comments to the Author**

Reviewer #2: All comments have been addressed

Reviewer #3: All comments have been addressed

2. Is the manuscript technically sound, and do the data support the conclusions?

Reviewer #2: Yes

Reviewer #3: Yes

3. Has the statistical analysis been performed appropriately and rigorously?

Reviewer #2: Yes

Reviewer #3: Yes

4. Have the authors made all data underlying the findings in their manuscript fully available?

Reviewer #2: Yes

Reviewer #3: Yes

5. Is the manuscript presented in an intelligible fashion and written in standard English?

Reviewer #2: Yes

Reviewer #3: Yes

Reviewer #2: I have reviewed the revised submission thoroughly and am satisfied with the amendments made. The authors have addressed the points raised in the initial review comprehensively.

Reviewer #3: (No Response)

**Do you want your identity to be public for this peer review?** For information about this choice, including consent withdrawal, please see our Privacy Policy

Reviewer #2: No

Reviewer #3: No

---

## [Editor Report · Acceptance letter]

PONE-D-25-18282R1

PLOS ONE

Dear Dr. Zaitsu,

I'm pleased to inform you that your manuscript has been deemed suitable for publication in PLOS ONE. Congratulations! Your manuscript is now being handed over to our production team.

Kind regards,

on behalf of

Dr. Emine Ozdemir Kacer

Academic Editor

PLOS ONE